# Virus Adaptation Following Experimental Infection of Chickens with a Domestic Duck Low Pathogenic Avian Influenza Isolate from the 2017 USA H7N9 Outbreak Identifies Polymorphic Mutations in Multiple Gene Segments

**DOI:** 10.3390/v13061166

**Published:** 2021-06-18

**Authors:** Klaudia Chrzastek, Karen Segovia, Mia Torchetti, Mary Lee Killian, Mary Pantin-Jackwood, Darrell R. Kapczynski

**Affiliations:** 1Southeast Poultry Research Laboratory, USA National Poultry Research Center, Agricultural Research Service, USA, Department of Agriculture, 934 College Station Road, Athens, GA 30605, USA; klaudia.chrzastek@gmail.com (K.C.); karens_122@outlook.com (K.S.); mary.pantin-jackwood@usda.gov (M.P.-J.); 2Diagnostic Virology Laboratory, National Veterinary Services Laboratories, U.S. Department of Agriculture, 1920 Dayton Ave, Ames, IA 50010, USA; mia.torchetti@usda.gov (M.T.); mary.l.killian@usda.gov (M.L.K.)

**Keywords:** avian influenza virus, whole genome sequencing, polymorphism, adaptation, transmissibility, duck, H7N9

## Abstract

In March 2017, highly pathogenic (HP) and low pathogenic (LP) avian influenza virus (AIV) subtype H7N9 were detected from poultry farms and backyard birds in several states in the southeast United States. Because interspecies transmission is a known mechanism for evolution of AIVs, we sought to characterize infection and transmission of a domestic duck-origin H7N9 LPAIV in chickens and genetically compare the viruses replicating in the chickens to the original H7N9 clinical field samples used as inoculum. The results of the experimental infection demonstrated virus replication and transmission in chickens, with overt clinical signs of disease and shedding through both oral and cloacal routes. Unexpectedly, higher levels of virus shedding were observed in some cloacal swabs. Next generation sequencing (NGS) analysis identified numerous non-synonymous mutations at the consensus level in the polymerase genes (i.e., PA, PB1, and PB2) and the hemagglutinin (HA) receptor binding site in viruses recovered from chickens, indicating possible virus adaptation in the new host. For comparison, NGS analysis of clinical samples obtained from duck specimen collected during the outbreak indicated three polymorphic sides in the M1 segment and a minor population of viruses carrying the D139N (21.4%) substitution in the NS1 segment. Interestingly, at consensus level, A/duck/Alabama (H7N9) had isoleucine at position 105 in NP protein, similar to HPAIV (H7N9) but not to LPAIV (H7N9) isolated from the same 2017 influenza outbreak in the US. Taken together, this work demonstrates that the H7N9 viruses could readily jump between avian species, which may have contributed to the evolution of the virus and its spread in the region.

## 1. Introduction

Avian influenza viruses can be divided into highly pathogenic avian influenza (HPAI) viruses and low pathogenic avian influenza (LPAI) viruses. Highly pathogenic avian influenza viruses are restricted to H5 and H7 subtypes and cause severe illness in gallinaceous species with high mortality rates, whereas LPAI viruses are typically maintained in wild aquatic birds and can be transmitted to domestic poultry species resulting in subclinical infection or mild respiratory disease. After circulation in poultry species, some H5 and H7 LPAI viruses can mutate to the HP genotype/phenotype through the accumulation of basic amino acids at the hemagglutinin (HA) cleavage site [1,2].

Highly pathogenic avian influenza outbreaks manifest as a direct result of two potential situations. In some cases, a flock becomes infected with a LPAI H5 or H7 virus that, if not detected and eradicated in time, can evolve into an HPAI virus. Alternatively, there can be a direct introduction from wild birds of an HPAI virus as was observed in the 2014–2015 outbreak of the H5N8/H5N2 HPAI in the United States [3,4]. This characteristic is unique to the Goose/Guangdong/96 H5Nx lineage of viruses. Most HPAI outbreaks (where the cause could be traced) can be attributed to within flock evolution, for example, outbreaks in the US in 1983 (H5N2), Mexico in 1994 (H5N2), Italy in 1999 (H7N1), and Canada in 2004 (H7N3) were all associated with the prior circulation of an LPAI progenitor virus [5,6,7]. Many HPAI outbreaks are relatively localized, even to the level of a single farm. The reasons for this are uncertain but likely reflect the burnout of the HPAI virus coupled with the obvious nature of the infection facilitating control (depopulation) measures. In addition, increased biosecurity in modern poultry facilities and lower fitness and/or transmissibility of the HPAI virus outside of a high-density poultry population contributes to its localization [8]. The lack of AI vaccine usage may also assist in this regard by not masking disease progression.

In 2017, outbreaks of the H7N9 LPAI virus (LPAIV) and HPAI virus (HPAIV) occurred at chicken poultry farms in Tennessee [4] and were found in mixed species backyard operations and markets in Alabama (AL), Kentucky (KY), and Georgia (GA), US. In total, between 4 March and 25 March, 2017, 14 premises were identified with confirmed H7N9 infections. Highly pathogenic avian influenza was confirmed on two premises, both in commercial broiler breeder flock, whereas the remaining premises had confirmed LPAI (six commercial premises and six backyard flocks). Molecular characterization of the 2017 US H7N9 viruses demonstrated that these viruses belong to North American wild bird lineage and are most closely related to a previously detected wild duck virus: A/blue-winged teal/Wyoming/AH0099021/2016 (H7N9) [9]. The pathogenesis of the A/chicken/Tennessee/17-007431-3/2017 H7N9 virus in commercial broiler breeders and SPF layer chickens was investigated; no transmission to contact-exposed birds was observed, and virus shedding was almost exclusively from the oropharyngeal route [3].

This study aimed to characterize the pathogenicity and transmissibility of a backyard duck-origin H7N9 LPAIV isolate in chickens, with downstream analysis of viral genetic changes to address the potential contribution of interhost transmissions in the spread of the virus during the outbreak. We also conducted NGS using RNA obtained directly (without chicken embryo passage) from pooled duck swab clinical specimens submitted during the H7N9 outbreak for comparison to what was obtained in the experimental study with the same H7N9 virus. The results demonstrate that experimental infection with the duck-origin H7N9 LPAIV resulted in all chickens becoming infected and transmission of the virus to naive birds.

## 2. Materials and Methods

### 2.1. Challenge Virus

An original isolate of A/duck/Alabama/2017 (H7N9) (dk/AL) LPAIV, kindly provided by USDA, NVSL, Ames, Iowa, was used as the challenge virus. The virus was propagated by one pass in specific pathogen-free (SPF) embryonated chicken eggs according to standard procedures [10]. This virus was isolated by NVSL from a pooled sample obtained from oropharyngeal swabs collected from backyard ducks in Alabama in 2017 during the outbreak. The experiment was performed in animal biosecurity level-3 enhanced (ABSL-3E) facilities in accordance with procedures approved by the US National Poultry Research Center (USNPRC, Athens, GA, USA, AUP# DRK FY2104-05).

### 2.2. Clinical Specimens Obtained during the H7N9 AIV Outbreak

The RNA extract obtained from an oropharyngeal, pooled swab sample of backyard ducks-origin specimen (17-008643-23) collected on 14 March, 2017 (Madison Co., Madison County, AL, USA), was kindly provided by the USA Department of Agriculture (Washington, DC, USA), National Veterinary Service laboratory (NVSL) Ames, IA, USA. This specimen was used for direct, whole-genome sequencing without the egg passage.

### 2.3. Animals and Housing

Specific pathogen-free (SPF) White Leghorn chickens were obtained from the Southeast Poultry Research Laboratory (SEPRL, Athens, GA, USA) flock. The birds were housed within the animal biosafety level-2 (ABSL-2) facilities before challenge. Serum samples were collected from all birds prior to challenge to ensure that the birds were serologically negative for AIV. Subsequently, the birds were transferred to ABSL-3E facilities and kept in negative-pressure HEPA-filtered isolators.

### 2.4. Experimental Design and Sampling

All procedures were performed according to the protocol approved by the USNPRC Institutional Animal Care and Use Committee (IACUC) at the U.S. National Poultry Research Center, Athens, GA, USA (USNPRC). At three weeks of age, the birds (*n* = 5) were challenged intranasally (10^6^ EID50 per bird) with dk/AL LPAIV. Control birds (*n* = 5) received PBS. Twenty-four hours post challenge, contact birds (*n* = 2) were introduce to each isolator. Birds had ad libitum access to feed and water throughout the experiment. Birds were monitored daily for clinical signs of disease. Oropharyngeal (OP) and cloacal (CL) swabs were collected daily from day 1 to day 8 post challenge (dpc) in 1.5 mL brain heart infusion media (Becton, Dickinson and Company, Sparks, MD, USA) with penicillin (2000 units/mL; Sigma Aldrich, St. Louis, MO, USA), gentamicin (200 μg/mL; Sigma Aldrich; St. Louis, MO, USA), and amphotericin B (5 μg/mL; Sigma Aldrich; St. Louis, MO, USA) and stored at −80 °C. Serum samples were taken at day 0 and at termination on 14 dpc. At this time, birds were euthanized.

### 2.5. Influenza Targeted Whole Genome Amplification

Viral RNA was extracted using the MagMAX AI/ND Viral RNA Isolation Kit (Ambion, Austin, TX, USA). For experimental chicken swab samples and challenge virus inoculum, one step RT-PCR was conducted with 20 μL of RNA template in a final reaction volume of 50 μL using OneTaq^®^ One-Step RT-PCR Kit (NEB, Ipswich, MA, USA) with the primers Optil-F1, 5′-GTTACGCGCCAGCAAAAGCAGG-3′, Optil-F2, 5′-GTTACGCGCCAGCGAAAG CAGG-3′, and Optil-R1, 5′-GTTACGCGCCAGTAGAAACAAGG-3′ as described previously [4]. The PCR cycling was performed as follows: 95 °C for 2 min, 42 °C for 60 min, 94 °C for 2 min, 5 cycles of 94 °C for 30 s, 44 °C for 30 s, and 68 °C for 3.5 min, followed by 26 cycles of 94 °C for 30 s, 57 °C for 30 s, 68 °C for 3.5 min with a final extension at 68 °C for 10 min.

### 2.6. Whole Genome Sequencing

The Nextera XT DNA Sample Preparation Kit (Illumina, San Diego, CA, USA) and 0.2 ng/μL (1 ng total) of dsDNA, obtained after targeted whole genome amplification (Section 2.4), were used in this study to generate multiplexed paired-end sequencing libraries, according to the manufacturer’s instructions as previously described [4]. Pooled libraries were sequenced on a 500-cycle MiSeq Reagent Kit v2 (Illumina) on an Illumina MiSeq (Illumina). We used MIRA assembler v3.4.1 [11] for de novo genome assembly of the clinical specimen obtained during the H7N9 outbreak and challenge virus inoculum (dk/AL H7N9) using an in-house pipeline on the GALAXY platform previously described [12,13], while the experimental swab samples from in vivo experiments were assembled by mapping sequencing reads against de novo assembled dk/AL H7N9 virus inoculum. Briefly, the quality of sequencing reads was assessed using FastQC v0.11.5. The reads were quality trimmed using a quality score of 30 or more, in addition to low-quality ends trimming and adapter removal using Trim Galore v0.5.0 (https://github.com/FelixKrueger/TrimGalore, accessed on 20 November 2020). The resulting contigs were quality assessed using QUAST v5.0.2 [14]. Reference-based orientation and scaffolding of the contigs produced by the assembler were performed using Scaffold_builder v2.2 [15]. Consensus sequences were recalled based on BWA-MEM [16] mapping of trimmed (but un-normalized) read data to the genome scaffold and parsing of the mpileup alignment. Subsequently, the clinical specimen sequencing reads were mapped to de novo assembled consensus to perform single-nucleotide polymorphism (SNIP) analysis within the sequences using Geneious 9.1.2 software [17]. All experimental swab sample sequences were mapped to de novo assembled challenge virus inoculum dk/AL LPAIV, and the SNIP analysis was conducted using Geneious 9.1.2 software. The polymorphism analysis was computed using a minimum sequencing depth of 1000 reads covering the base and a minimum variant frequency of 5%. The maximum *p*-value of 10^−6^ and minimum strand-bias *p*-value of 10^−5^ when exceeding a 65% bias were used to call the variants with an average quality > 35.

### 2.7. Viral RNA Quantification in Experimental Swabs

Viral RNA was extracted using Trizol LS reagent (Invitrogen, Carlsbad, CA, USA) and the MagMAX AI/ND Viral RNA Isolation Kit (Ambion, Austin, TX, USA). Quantitative real-time RT-PCR (RRT-PCR) was performed using primers and a probe specific for the type A avian influenza (AI) matrix gene (Spackman, 2002 #104). The reactions were conducted using AgPath-ID one-step RT-PCR Kit (Ambion) and the ABI 7500 Fast Real-Time PCR system (Applied Biosystems, Carlsbad, CA, USA). For virus quantification, a standard curve was established with the RNA extracted from dilutions of the same titrated stock of the challenge virus. Cycle threshold (CT) values of each viral dilution were plotted against viral titers. The resulting standard curve had a high correlation coefficient (*R*^2^ > 0.99), and it was used to convert CT values to EID50/mL [2,18]. The lower limit of detection was 0.9 × log10 EID50 per mL.

### 2.8. Serology

Hemagglutinin inhibition (HI) assays were carried out using inactivated dk/AL (H7N9) antigen. Hemagglutinin inhibition assays were performed according to standard procedures [4]. The HI titer was expressed as log2 geometric mean titer (GMT). Samples with titers at or above 3 × log2 GMT were considered positive.

## 3. Results

### 3.1. Clinical Signs of Experimental Infection of Chickens with an H7N9 LPAIV Duck Isolate

All SPF layer chickens that were challenged with A/duck/Alabama/2017 (H7N9) (dk/AL) LPAIV survived the challenge. All the sham birds remained clinically healthy for the duration of the experiment. The challenged birds showed mild clinical signs including swollen and watery eyes, and some birds presented green diarrhea. The first clinical signs were observed at 2 dpc. One of the directly inoculated birds (#584) exhibited more severe clinical signs, including nasal discharge, bleeding follicular conjunctivitis, and bloody diarrhea. This bird recovered at 6 dpc and remained clinically healthy until the end of the experiment. Notably, conjunctivitis, tearing of the eyes, nasal discharge, and diarrhea were also observed in the two contact birds at day 2 post introduction to infected birds and continued for 2 days.

### 3.2. Virus Shedding

Quantitation of viral shed was performed by qRT-PCR using extrapolation of a standard curve generated with the challenge viruses via virus isolation and titration. Oral and cloacal swab samples from direct inoculated birds were collected daily from day 1 to day 8 post challenge (dpc). To evaluate transmission, two naive contact birds were introduced in the isolator with inoculated birds 24 h pc and sampled until day 7 post exposure (dpe). All directly infected chickens shed the virus through the OP route, starting at 1 dpc (Figure 1A) with an average titer of 10^3^ EID50/mL.

Birds shed the virus orally until day 6 post challenge (dpc). The peaks of viral shedding were at 2 and 5 dpc. Similarly, contact birds shed the virus orally from day 2 post exposure (Figure 1C) but at lower titers and for a shorter period of time (4 days vs. 6 days) compared to directly inoculated birds.

Three out of five directly infected birds shed the virus through the cloaca (Figure 1B). Surprisingly, the viral titers observed from the cloaca samples were, in some cases, higher than seen by the oral route (Figure 1). The highest level of viral shedding was demonstrated from the cloacal between 4 and 6 dpc. At 7 dpc, two birds shed a virus at titers above 10^3^ EID50. One bird shed the virus through the cloaca at day 8 post challenge. In addition, the two contact birds shed the virus cloacally starting from 3 dpe (Figure 1D) and at similar titers as the directly inoculated chickens. The cloacal viral titers were higher than that found in oral swabs (Figure 1). At day 5 post exposure, both contact birds shed the virus at titers above 10^4^ EID50/mL, and both birds still shed a virus at day 7 post exposure (Figure 1D).

### 3.3. HI Titers

Serum samples were obtained for the determination of antibody titers against the corresponding challenge virus (Appendix A). All directly infected chickens seroconverted (>7 × log2). The average HI titer in directly infected chickens was 8.5 × log2. Both contact birds seroconverted with an average titer of 6.5 × log2.

### 3.4. Polymorphism Analysis

The A/duck/AL/2017 (dk/AL) H7N9 challenge virus sequenced in this study had no polymorphism sites at a minimum of 5% frequency within the segments and, thus, no mix of viruses was present in the viral inoculum.

A direct genome comparison between the LP-17-008643-2 clinical duck specimen obtained during the outbreak and A/duck/AL/2017 H7N9 virus isolate was possible only for the NP, M, and NS viral segments, as no other segments of the LP-17-008643-2 clinical sample were recovered in this study. The results in the duck clinical specimen demonstrated two synonymous changes in the M1 segment (at position 25, T to C and 127, A to G) and a mixed population of viruses at position 139 in the NS protein with a majority of viruses carrying 139D and 21.4% carrying 139N. This subpopulation of viruses was not present in the A/duck/AL/2017 virus isolate used as a challenge virus in this study. No polymorphism was found in the NP segment of the LP-17-008643-2 clinical sample. The homology analysis of NP, M, and NS consensus sequences assembled from the duck sample clinical specimen showed high similarity with the Influenza A virus A/blue-winged teal/Texas/AI13-3557/2013(H4N6)) (99.5%), Influenza A virus A/blue-winged teal/Texas/UGAI15-6706/2015(H1N1) (99.7% similarity), and Influenza A virus A/mallard/Alaska/16-029597-2/2016(H5N2)) gene (99.5%), respectively, while the similarity score to A/duck/Alabama/2017 (H7N9) viral isolate was below 98%. Interestingly, the duck H7N9 LPAIV sample carried the same M105I substitution in the NP segment as the H7N9 HPAIV chicken isolates, contrary to chicken H7N9 LPAIVs (Appendix A).

The chicken experimental swab samples were aligned to de novo generated consensus sequences of the dk/AL virus inoculum, and the polymorphism analysis was performed in accordance with our quality requirements (Section 2). The non-synonymous changes (>5% level) in viral segments recovered from (i) directly infected chickens, excluding chicken #584 (Table 1); (ii) contact chickens (Table 2); (iii) directly inoculated chicken #584 (Table 3) are presented separately. Directly inoculated with dk/AL, chicken #584 exhibited more severe clinical signs compared to other birds (Section 3.1) and, thus, we wanted to evaluate whether there were minor population of viruses that potentially evolved over the time of infection.

All RNA extracts tested in this study were LPAIV based on amino acid sequence at the HA proteolytic cleavage site (PENPKTR/G). At the consensus level (above 50%), there were no amino acid changes in any of the viral segments recovered from directly challenged chickens at 4 or 6 dpc (Table 1). However, a diverse viral population beyond the consensus level with a large number of polymorphic sites in the polymerase complex genes was observed (Table 1). In total, 19 non-synonymous and three synonymous changes in the PA segment, 16 non-synonymous and seven synonymous changes in PB2, and seven non-synonymous and four synonymous changes in PB1 (Table 1 and Appendix A) were observed. In addition, two non-synonymous changes in NP (N450D and A336T) and two non-synonymous changes in NS (K27M and M79I) were found (Table 1). Both M and NA segments were not polymorphic.

Next, we assessed genome changes of the dk/AL/17 (H7N9) viruses recovered from contact birds (Table 2). The non-synonymous changes in PB2, PB1, PA, and HA nucleotide sequences of functional or structural importance are shown in Table 2. The majority of the PA polymorphic sites were found in the C-terminus region that interacts with the N-terminal region of PB1 (Table 2). Interestingly, 30.5% of viruses caried substitution S150L (S143L, H3 numbering) in the HA segment. No substitutions were found in the NP, M, or NS viral segments.

Because one of the directly inoculated birds (#584) exhibited more severe clinical signs compared to other challenge chickens (Section 3.1.), NGS with polymorphism analysis was applied to RNA extracts recovered from swab samples taken from this bird at different time points to assess genome changes acquired. At the consensus level (above 50%), the following substitutions were observed: V203I in PB2 at 3 dpc (D3CL) and A337T in NP at 7 dpc (D7CL) (Table 3). Furthermore, minor populations of viruses carried substitutions (>5%) including T530A at D1OP and V109A and N639S at D3 CL in PB2; L42F at D1OP and E457K at D3CL in PA; M12V, T321A, and F409Y at D1OP and A169V and K337N at D3CL in HA; G356E at D1OP, V67A at D3CL in NP; S22P in M2 and F251S in M1 at D1OP (Table 3). Interestingly, K337N (K328N H3 numbering) substitution in the HA segment was carried by almost 40% of viruses while A169V (A160V H3 numbering) (13% of viruses) recovered from clinically ill chicken (Table 3).

## 4. Discussion

In March 2017, H7N9 LPAI and HPAI cases occurred in chickens in Tennessee and subsequently in Alabama, Kentucky, and Georgia, USA. Phylogenetic network analysis of LP and HP H7N9 TN and AL isolates suggest that the virus circulated undetected, and the mutation from an LPAI virus to an HPAI virus occurred in poultry [9]. In order to address the possible role of domestic ducks in the spread of H7N9, the pathogenicity and transmissibility of a domestic duck-origin LPAIV, A/duck/Alabama/17-008643-2/2017(H7N9), were examined in SPF chickens. The results demonstrate that the LPAIV H7N9 duck isolate readily infected directly challenged birds, transmitted to contact-exposed birds, and caused clinical signs, which indicates that this virus is well adapted to chickens. Interestingly, a previous study tested the ability of a chicken-origin H7N9 LPAI isolate from the same outbreak and found that the virus could not transmit to contact-exposed chickens [3]. This may be due to the lower level of virus replication in those birds. Pantin-Jackwood et al. [28] have shown that LP and HPAIV from the US 2016 H7N8 in turkeys could infect mallards and turkeys and transmit to contacts. However, when chickens were inoculated with the same viruses neither could be transmitted to contact-exposed chicken. Therefore, there is a fair degree of variability and unknown factors that contribute to transmission in experimental settings.

To assess whether the increased fitness of A/duck/Alabama/17-008643-2/2017(H7N9) in chickens relates to viral genome changes, we sequenced the RNA extracts recovered from swab samples. A large number (*N* = 42) of non-synonymous changes were found in the replication machinery genes (PB2, PB1, and PA segments), some of them being found in positions of functional or structural importance. The changes found in PB2-, PB1-, and PA-binding sites could improve the interplay of polymerase subunits in a new host environment. Although the majority of SNPs found appeared below consensus level, it is known that diverse sequences of viral populations within individual hosts are the starting material for selection and subsequent evolution of RNA viruses [29,30]. For instance, a minor population of viruses that carried mutations at positions 702 or 649 in PB2, 678 and 695 in PB1, and 85 in PA appeared in cloacal RNA extracts recovered from contact birds that were previously described to be responsible for increased polymerase activity and interspecies transmission [8,19,20,21,22,23,24,25]. Youk et al. [31] have found PB2 change V649I in the genome of H5N2 HPAIV that contribute to increased virus fitness in chickens. In addition to the polymerase segments, we found that 30.5% of viruses recovered from RNA extracts shed by contact birds carried a substitution at position 143 (H3 numbering) in the hemagglutinin (HA). Masuda et al. [26] have shown that the position 143 in HA, which is located in the receptor binding site, is linked to the viral recognition of 5-N-glycolylneuraminic acid (Neu5Gc). Furthermore, position 143 is located in an HA antigenic site, and amino acid changes in natural isolates are principally chosen for changeable substitutions in this site such as P143S in H3HA protein during evolution [27]. We hypothesize that the molecular changes in the polymerase genes of the LPAIV H7N9 duck-origin virus, once inoculated into chickens, allowed for increased viral replication which could facilitate the spread of the virus. Furthermore, additional changes in the viral genome, especially in the receptor binding sites of the HA could facilitate the entry of the virus into cells.

Experimental infection of chickens with dk/AL/17(H7N9) resulted in higher titers and duration of shedding compared to chicken-origin H7N9 LPAIV and HPAIV from the same outbreak when examined in chickens [3]. The duck-origin virus examined in this study was also shed via the cloacal route, which was only observed with the HPAIV in the previous study [3]. Cloacal shedding in chickens is rarely observed with LPAIVs [3,32], indicating that the viral shedding observed in some of the chickens inoculated with dk/AL/17 could be caused by the changes the virus underwent as it adapted to ducks, where intestinal replication and cloacal viral shedding is common. Viral shedding by both routes, OP and CL, could also increase the amount of virus in the environment and explain the transmission of this virus to contacts. Sequence analysis of RNA extracts obtained at day 3 post challenge from that bird showing the more severe clinical signs found a minor population of viruses shed thorough cloaca carrying the substitution K337N (K328N H3 numbering) (almost 40% of viruses) in the HA cleavage site, and A169V (A160V H3 numbering) (13% of viruses) in the HA receptor binding site. Gu et al. [33] have shown that T160A HA substitution in a H5N1 clade 2.3.4 virus resulted in the loss of a glycosylation site at 158N and led to enhanced binding specificity for human-type receptors and transmissibility among guinea pigs. To determine whether residue at position 160 (H3 numbering) is conserved among H7N9 avian influenza viruses, we looked at the 1979 H7 sequences from 2008, available in the influenza database (fludb.org, accessed on 20 March 2020), and perform sequence variation analysis (fludb.org, internal pipeline). The majority of H7N9 AIVs carry Ala at position 160 in HA (*N* = 1900), followed by Thr (*N* = 49). Interestingly, there were 29 natural H7N9 viruses that carried Val at position 160 in the HA molecule. Two recent isolates, A/blue-winged teal/Louisiana/UGAI15-2471/2015(H7N3) and A/duck/Cambodia/b0120501/2017(H7N3) carried valine at position 160 in their hemagglutinin segment. Interestingly, the Cambodia 2017 H7N3 duck strain caused high mortality (93%) in Khaki Campbell ducks (Anas platyrhynchos domesticus) in the Kampong Thom Province [34]. The question whether this position could serve as an important molecular marker for assessing increased pathogenicity and/or binding affinity of H7 viruses to chicken/duck cells is open and needs to be further investigated.

The sequence analysis of RNA extracted from the backyard duck clinical specimen, showed a mixed population of viruses at position 139 in NS1 protein, (139N 22%, and 139D, 88% of viruses), while the egg propagated, the inoculum virus carried only aspartic acid (D) at the same position. It was previously demonstrated that amino acid substitutions in NS1 protein, including N139D of A/quail/Hong Kong/G1/97 (HK/97) H9N2, resulted in binding to the 30 kDa subunit of the cleavage and polyadenylation specificity factor (CPSF30), and, in consequence, inhibition of host immune response [35]. Wild ducks are known as a natural reservoir of LPAIV strains [32,36] and can carry viruses with no apparent disease signs. Wallensten et al. [37] have shown that all subtypes of HA (H1–H12) and all subtypes of NA can be found in 55 different combinations in mallard ducks. The sequence analysis of the NP, M, and NS gene segments of the backyard duck original clinical specimen presented in this study showed close relatedness to the H4N6 and H1N1 blue-winged teal Texas (2015) and mallard H5N2 Alaska (2016) isolates. Interestingly, the H7N9 LPAIV duck isolate along with the H7N9 HPAIV chicken isolates carried M105I mutation in nucleoprotein genes. Tada et al. [38] have shown that a single amino acid substitution to valine at position 105 in NP (105V) in the H5N1 A/duck/Yokohama/aq10/2003 virus was responsible for the increased pathogenicity of a duck isolate in chicken cells. To determine if this residue is variable among avian (turkey, chicken, mallard) and human isolates, we analyzed the 2346 NP sequences of 2025 avian and 321 human H5Nx influenza viruses isolated from 2000, available in influenza database (fludb.org, internal pipeline). The sequence variation (fludb.org, internal pipeline) indicated that 72% and 62% of mallard and turkey isolates carry methionine (M) at position 105 in NP, respectively. In contrast almost 84% and 81% of chicken and human isolates carry 105V. A 105I substitution was carried by 39% of turkeys, 10.4% mallards, 6% chicken, and less than 1% human H5Nx AIVs. A similar search of NP sequences demonstrated that 66% of mallard isolates carried 105M, whereas AIV isolated from chicken, turkey, or human species carried the substitution M105V in NP (93%, 89%, and 85%, respectively). This sequence analysis data might support the idea that the position 105 in NP may be a determinant for host adaptation or/and increased pathogenicity. Youk et al. [39] demonstrated that the Mexican H7N3 HPAI viruses, after circulating in chicken populations for several years, retained high pathogenicity to chickens but decreased fitness in mallards. Some of the changes observed in the viral genomes of Mexican H7N3 between 2012 and 2016 were associated with a point mutation at A125T in HA, NP M105V, and NP S377N [39]. In addition, M105V and M105I were also found as the H5N2 HPAIV Clade 2.3.4.4 adapted to poultry in the USA [31].

To conclude, the dk/AL/17 (H7N9) LPAIV readily infected and transmitted to contact-exposed birds and was shed through both oral and cloacal routes. Interestingly, LPAI H7N9 virus isolated from the same US outbreak from chicken A/chicken/Tennessee/17-007431-3/2017 (GenBank accession numbers KY818816-KY818823) with almost identical genome sequence did not transmit to contact control birds and was not detected from cloacal swabs. The observation of higher shedding in cloacal swabs appears to be unique and requires further work to decipher. The wide range of polymorphisms identified in RNA extracts recovered from experimentally infected SPF layer hens is an apparent attempt at species adaption.

## Figures and Tables

**Figure 1 viruses-13-01166-f001:**
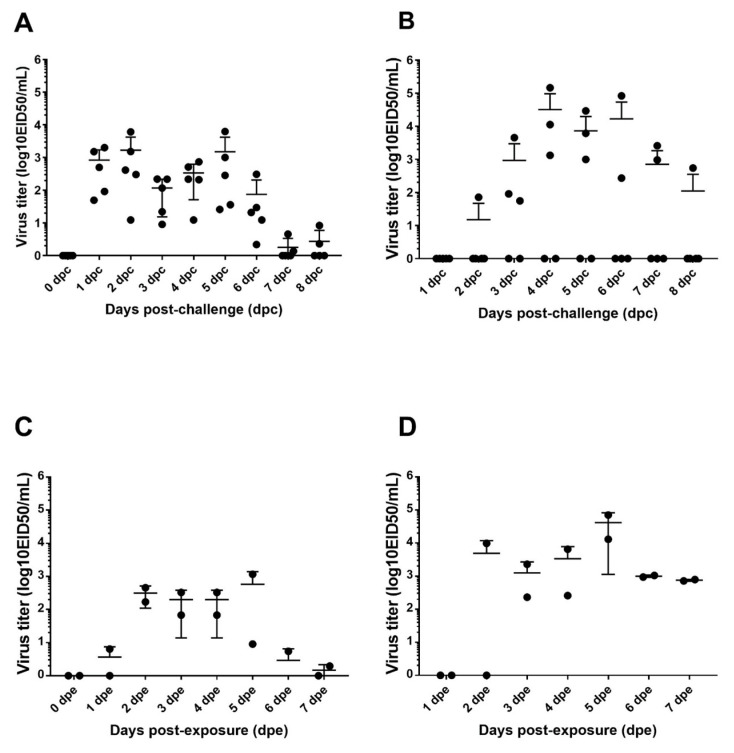
Viral titers from oral (**A**,**C**) and cloacal (**B**,**D**) swabs from day 0 to day 8 post challenge (dpc) or day 7 post exposure (dpe). Chickens were challenged with the A/duck/AL/2017 (H7N9) LPAIV virus at 3 weeks of age, and the swabs were collected daily. The contacts birds were introduced 24 h post challenge.

**Table 1 viruses-13-01166-t001:** The non-synonymous changes in the A/duck/AL/2017 (H7N9) LPAIV viral genome segments recovered from swab samples at day 4 and 6 post challenge from directly infected SPF chickens. Chickens were challenged intranasally with 10^6^ EID50 of virus inoculum per bird at three weeks of age. The frequencies of viral variants were computed using a minimum sequence coverage of 1000 reads covering the base and a minimum variant frequency of 5%. The maximum *p*-value of 10^−6^ and a minimum strand-bias *p*-value of 10^−5^ when exceeding a 65% bias were used to call the variants with an average quality > 35.

Day Post Challenge	Segment	CDS Codon Number	Amino Acid Change	Variant Frequency	Sequencing Depth
Day 4	PB2	74	G −> E	20	5767
		77	L −> P	14.7	4496
		647	I −> R	5.7	3797
		649	V −> G	25.7	4433
		653	S −> T	5.6	5385
		656	F −> L	5.1	6358
		657	N −> K	7.5	6864
		659	N −> K	5.9	7711
		661	A −> T	7.5	8439
		702	K −> E	5.1	22,021
		703	R −> K	5.5	23,430
		717	A −> T	10.3	31,662
	PB1	695	L −> I	33.3	1627
	PA	89	T −> P	10	3243
		91	V −> G	8.9	2719
		569	G −> R	43.4	1410
		701	L −> S	8.1	5705
	NS	79	M −> I	17.6	17,973
Day 6	PB2	66	M −> L	6	2393
		94	L −> V	34.1	1026
		518	V −> I	9.8	1585
		686	V −> E	5.5	9359
	PB1	625	C −> G	33	2676
		631	F −> S	7.2	2757
		656	E −> K	5.1	4535
		678	S −> R	5.3	7818
		679	Q −> K	6.6	8010
		680	R −> K	7.2	8109
	PA	23	E −> G	5.1	9793
		25	G −> W	6.5	10,646
		84	R −> H	5.6	12,405
		85	T −> N	5.2	12,364
		116	R −> H	5.5	4283
		142	K −> T	13.6	1059
		142	K −> R	6.4	1059
		548	M −> I	7.1	1010
		554	I −> R	5	1608
		556	Q −> H	6.9	2004
		559	R −> G	7.5	2226
		561	M −> L	9.1	2519
		581	M −> I	7.1	5204
		616	S −> A	5.6	12,713
		652	S −> F	5.3	26,340
	NP	336	A −> T	49	18,924
		450	N −> D	11.9	11,794
	NS	27	L −> M	9.4	20,652
		79	M −> I	17.6	17,973

**Table 2 viruses-13-01166-t002:** Molecular changes in the A/duck/AL/2017 (H7N9) LPAIV viral genome assembled from RNA extracts recovered from contact chickens at day 4 and day 6 post exposure from cloacal swab samples. Contact chickens were transferred into directly challenged chickens at 24 h post challenge. The frequencies of variants were computed using a minimum sequence coverage of 500 reads covering the base and a minimum variant frequency of 5%. The maximum *p*-value of 10^−6^ and a minimum strand-bias *p*-value of 10^−5^ when exceeding a 65% bias were used to call the variants with an average quality > 35.

Gene	Position	Variant Frequency (%)	Depth of Coverage	Category ^1^	Description
PB2	K702E	5.1	22,021	S, F	The presence of K702R substitution enhances viral transmission to humans [8,19,20]
	R703K	5.5	23,430	S	
PB1	S678R	5.3	7818	F	A determinant of host range. The PB1 13P and 678N, together with PB2 701N and 714R, PA 615N, and NP 319K cause a dramatic increase in polymerase activity and confer adaptation of AIV to mammalian hosts [21]
	L695I	33.3	1627	F	PB2 binding site. This interface is crucial for the regulation of overall enzyme activity. The C-terminal three helix bundle of PB1 binds to 1–37 and 1–86 fragments on the N-terminus of PB2 [22,23]
PA	T85N	5.2	12,364	F	Residues responsible for enhanced polymerase activity in mammalian cells [24,25]
	V91G	8.9	2719		
	M548I	7.1	1010	F	PB1 binding side
	I554R	5	1608		
	Q556H	6.9	2004		
	R559G	7.5	2226		
	M561L	9.1	2519		
	G569R	43.4	1410		
	M581I	7.1	5204		
	S616A	5.6	12,713		
	S652F	5.3	26,340		
	L701S	8.1	5705	F	The C-terminus of PA interacts with the N-terminal region of PB1 (residues 1–25). This subunit interface complex is essential for initiation of transcription
HA	S150L (S143L)	30.5	3470	F	A receptor-binding side [26,27]

^1^ S = structural; F = function.

**Table 3 viruses-13-01166-t003:** Polymorphisms found in the A/duck/AL/2017 (H7N9) viral genome recovered from SPF layer chickens (#584) that developed severe clinical disease. Chickens were challenged with A/duck/AL/2017 (H7N9) at three weeks of age. At day 2 post challenge, one directly infected chicken (#584) exhibited more severe clinical signs, including nasal discharge, bleeding follicular conjunctivitis, and bloody diarrhea, compared to other directly infected ones. This bird recovered at 6 dpc and remained clinically healthy until the end of experiment. The frequencies of variants were computed using a minimum sequence coverage of 1000 reads per base and a minimum variant frequency of 5%. The maximum *p*-value of 10–6 and a minimum strand-bias *p*-value of 10–5 when exceeding a 65% bias were used to call the variants with an average quality > 35.

Sample ^1^	AIV Gene Segment							
	PB2	PB1	PA	HA	NP	NA	M	NS
D1-OP	T530A (12.6) ^3^	NC ^2^	L42F (7.1)	M12V (6.4), T321A (16.9), F409Y (9.7)	G356E (7.5)	NC	S22P M2/F251S M1 (41.3)	NC
D3-CL	V109A (9.2), V203I (100), N639S (25.9)	NC	E457K (10.5)	A169V (13.3), K337N (39.8)	V67A (5.2)	NC	NC	NC
D4-OP	N/A ^4^	N/A	N/A	N/A	F39V (9.2)	N/A	N/A	NC
D4-CL	N/A	N/A	N/A	N/A	NC	N/A	NC	T90A (9.8) NEP
D6-CL	N/A	N/A	N/A	N/A	N/A	N/A	N/A	V111M (99.9) NS1
D7-CL	N/A	N/A	N/A	L537P (5.3)	A337T (99.1), E454G (8.7)	N/A	G89S M2 (18.0)	N/A

^1^ OP = oral; CL = cloacal. ^2^ NC = no changes. ^3^ (#) = Percentage of viral population containing polymorphism. ^4^ N/A = no data.

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
