# Peer review of "Virus Adaptation Following Experimental Infection of Chickens with a Domestic Duck Low Pathogenic Avian Influenza Isolate from the 2017 USA H7N9 Outbreak Identifies Polymorphic Mutations in Multiple Gene Segments"

_viruses, 2021, doi:10.3390/v13061166_

Round 1
Reviewer 1 Report
This is the review of the manuscript entitled:
“Virus adaptation following experimental infection of chickens 2 with a domestic duck low pathogenic avian influenza isolate 3 from the 2017 U.S. H7N9 outbreak identifies polymorphic mutations in multiple gene segments”
In this study, the authors analyzed the replication and transmission capacities of a H7N9 LPAIV after an experimental protocol consisting in the infection of White Leghorn chickens followed by the introduction of contact birds. A genetic analysis of the viral strain(s) was then conducted using Illumina sequencing in order to identify some genetic markers that could be involved in host adaptation.
This study was correctly designed and it is obvious that the authors are experts in the field of avian influenza and NGS analysis, the terms and the analyses are quite well relevant. It is for example very much appreciated that the authors worked directly from RNA without chicken embryo-passage due to the technical difficulties of such approaches.
The manuscript is clear and the discussion is remarkably supported by the literature.
I don’t have remarks concerning the study design or results interpretation. However, some information about NGS are missing.
- My comments to the authors are as follows:
Line 134: Why did the authors applied two types of analysis? I mean de novo for the field sample versus mapping for the experimental samples. Please clarify this point.
Line 144: Why did the authors use Geneious for SNP analysis whereas the previous steps were performed with open-source tools? I was wondering if a tool such as iVar could be relevant (Grubaugh 2019).
Results section: raw data information is missing: sequencing yield values for the whole run and for each sample. Mean read size is also missing. The authors are also encouraged to provide a clear table (suppl. data) showing the sequencing coverage (%) and the sequencing depth (X) of each gene of each sample.
Line 213: Why did the authors set a frequency at 5%? Illumina quality could allow a lower threshold.
In the discussion, the authors properly commented the role of each detected mutation. However, the authors are encouraged to discuss the strength and the weakness of the NGS approach they applied. Long read technology could for example help to connect different SNP initially present on the same RNA segment/allele.
The authors are encouraged to discuss the lack of data for some dk/AL isolate segments.
Aerosol and environmental sampling could also be discussed considering the unmissable role of aerosols in respiratory diseases transmission.
- I would also suggest some minor remarks:
2.6 section: Add references for MIRA (Chevreux), QUAST (Guverich), Scaffold_bluider (Silva), BWA-MEM (Li 2013).
Line 22: …Next generation sequencing…
Line 127: Specify that dsDNA was produced in 2.4 section.
Line 148: …sequencing depth of 1000 reads covering the base…
Line 193: …until day 6 post-challenge.
Line 194: …day 2 post-exposure…
Line 201: …103…
Line 221: Please substitute quasispecies for population,subpopulations or variant.
Line 229: …NP segment…
Line 230: …chicken H7N9…
Line 239: …infection…
Table 1: Chickens were challenged…
Table 1/first line: Sequencing depth or depth of coverage instead of coverage.
Reviewer 2 Report
This paper reviews the sequence outcome from a single passage in recipient chickens of a LPAI of H7N9. The significance of this work in this field is improtant to understand the adaptation of viruses to interacting species indicating the potential of such viruses to infect production animals and exposure to humans. I beleive that the title of the manuscript is misleading and should be altered. The authors should comment why thsi aprticular virus was selected and should analyze further viruses of H7N9 LPAI. The findings of the work presented suggests that by and large this virus has already adapted to replication in the recipient animals with high levels of replication and quasispecies generation. These findings adds little to the field and do not give the reader an indication as to what has occurred with this virus to enable theincreased the level of replication or adaptation to chickens.
The authors could improve the manuscript by doing multiple rounds of contatc infection and establishing the species that becomes dominant. Following this analysis, investigation into the functional aspect of the dominant genotypes that arise from multiple passage. For example, transcriptional analysis of the dominant polymerase mutations indicating increased activity to HA substitutions to the original strain and comparing in eggs the level of replication. Alternatively, the manusript would benefit from the infection/transmission of a range of other H7N9 viruses in their system.
Minor grammatical errors occur in the results section:
"One of directly" – should read one of the directly
"Because one of directly" – should read because one of the directly
Round 2
Reviewer 2 Report
I dont think this work is of any great significance since while the authors have sequence the passage of this virus from one species to the next and noted some changes, there is little to no functional data to interpret threse changes.